# Association between early gestation passive smoke exposure and neonatal size among self-reported non-smoking women by race/ethnicity: A cohort study

Melissa M. Amyx[1], Rajeshwari Sundaram[2], Germaine M. Buck Louis[3], Nicole M. Gerlanc[4], Alaina M. Bever[1], Kurunthachalam Kannan[5], Morgan Robinson[5], Melissa M. Smarr[6], Dian He[4], Fasil Tekola-Ayele[1], Cuilin Zhang[1], Katherine L. Grantz[1]*

1 Epidemiology Branch, Division of Intramural Population Health Research, *Eunice Kennedy Shriver* National Institute of Child Health and Human Development, Bethesda, MD, United States of America, 2 Biostatistics and Bioinformatics Branch, Division of Intramural Population Health Research, *Eunice Kennedy Shriver* National Institute of Child Health and Human Development, Bethesda, MD, United States of America, 3 Office of the Dean, College of Health and Human Services, George Mason University, Fairfax, VA, United States of America, 4 The Prospective Group, Inc., Fairfax, VA, United States of America, 5 Department of Pediatrics and Department of Environmental Medicine, New York University School of Medicine, New York, NY, United States of America, 6 Department of Environmental Health, Rollins School of Public Health, Emory University, Atlanta, GA, United States of America

* katherine.grantz@nih.gov

**Data Availability Statement:** Data and accompanying files can be requested from https://brads.nichd.nih.gov/; to gain access, data

## Abstract

Understanding implications of passive smoke exposure during pregnancy is an important public health issue under the Developmental Origins of Health and Disease paradigm. In a prospective cohort of low-risk non-smoking pregnant women (NICHD Fetal Growth Studies —Singletons, 2009–2013, N = 2055), the association between first trimester passive smoke exposure and neonatal size was assessed by race/ethnicity. Plasma biomarker concentrations (cotinine, nicotine) assessed passive smoke exposure. Neonatal anthropometric measures included weight, 8 non-skeletal, and 2 skeletal measures. Linear regression evaluated associations between continuous biomarker concentrations and neonatal anthropometric measures by race/ethnicity. Cotinine concentrations were low and the percent above limit of quantification varied by maternal race/ethnicity (10% Whites; 14% Asians; 15% Hispanics; 49% Blacks). The association between cotinine concentration and infant weight differed by race/ethnicity (P$_{interaction}$ = 0.034); compared to women of the same race/ethnicity, per 1 log-unit increase in cotinine, weight increased 48g (95%CI -44, 139) in White and 51g (95%CI -81, 183) in Hispanic women, but decreased -90g (95%CI -490, 309) in Asian and -93g (95%CI -151, -35) in Black women. Consistent racial/ethnic differences and patterns were found for associations between biomarker concentrations and multiple non-skeletal measures for White and Black women (P$_{interaction}$<0.1). Among Black women, an inverse association between cotinine concentration and head circumference was observed (−0.20g; 95%CI −0.38, −0.02). Associations between plasma cotinine concentration and neonatal size differed by maternal race/ethnicity, with increasing concentrations associated with decreasing infant size among Black women, who had the greatest biomarker

requestors will need to sign a data user agreement in accordance with NICHD Intramural Policy.

**Funding:** This work was supported by the National Institutes of Health's Intramural Research Program at the Eunice Kennedy Shriver National Institute of Child Health and Human Development (contract numbers HHSN275200800013C; HHSN 275200800002I; HHSN27500006; HHSN275200800003IC; HHSN275200800014C; HHSN275200800012C; HHSN275200800028C; HHSN275201000009C and HHSN27500008). One or more of the authors are employed by a commercial company: The Prospective Group, Inc. The funder provided support in the form of salaries for authors NG and DH, but did not have any additional role in the study design, data collection and analysis, decision to publish, or preparation of the manuscript. The specific roles of these authors are articulated in the 'author contributions' section.

**Competing interests:** NG and DH are employed by a commercial company: The Prospective Group. This does not alter our adherence to PLOS ONE policies on sharing data and materials. The authors have declared that no additional competing interests exist.

concentrations. Public health campaigns should advocate for reducing pregnancy exposure, particularly for vulnerable populations.

## Introduction

Previous research has established that cigarette smoking negatively affects fetal growth [1, 2] and birth size (for example, birthweight is reduced approximately 150-300g among women continuing to smoke during pregnancy) [1]. Although public health campaigns and regulation of clean-indoor air have reduced smoking prevalence and second-hand/passive smoke exposure [3], an estimated 15% of US non-smoking women are regularly exposed to passive smoking [4]; thus, a similar percentage of non-smoking pregnant women and their fetuses [5] may nonetheless be exposed to harmful chemicals contained in cigarette smoke, which even at low levels of exposure negatively impacts fetal growth [6]. Nicotine, and its metabolite cotinine, serves as a biomarker for measuring tobacco smoking exposure, both active and passive, and can readily cross the placenta [7, 8]. Further, nicotine has been directly implicated as having deleterious effects on fetal growth [8].

Additionally, differences in exposure to passive smoking and nicotine and cotinine metabolism have been observed among racial/ethnic groups [5, 9]. Specifically, slower nicotine metabolism has been observed among Black and Asian women compared to other racial/ethnic groups, resulting in higher plasma biomarker concentrations at the same level of exposure [3, 5, 10]. Given these differences, and that racial/ethnic differences in fetal growth have been reported [11], it is unclear whether associations between passive smoking exposure and fetal growth are consistent across racial/ethnic groups.

As prior studies largely focused on the impact of either active or passive smoking in association with birthweight, elucidating the implications of passive smoking exposure on specific neonatal anthropometric measures and whether there are differences by race/ethnicity is informative to public health campaigns. Therefore, our objective was to determine if the relationship of plasma concentrations of nicotine and cotinine with neonatal anthropometry differed by race/ethnicity among nonsmoking pregnant women (whose biomarker levels would indicate passive smoke exposure) with low-risk antenatal profiles.

## Materials and methods

### Study design and participants

The NICHD Fetal Growth Studies-Singletons was a prospective pregnancy cohort study conducted between July 2009 and January 2013 at 12 US clinical sites (ClinicalTrials.gov identifier: NCT00912132; additional information available at: https://www.nichd.nih.gov/about/org/diphr/officebranch/eb/fetal-growth-stud). Participants were recruited between 8 weeks 0 days (8w0d) and 13w6d gestation. Study participants included women aged 18–40 years with low risk antenatal profiles (e.g., non-smokers, BMI 19.0–29.9 kg/m$^2$) with viable, spontaneously conceived, non-anomalous singleton pregnancies, a low-risk medical and obstetrical history, and planning to deliver at a study hospital [11]. All participants provided written informed consent. Full human subjects' approval was obtained from all participating clinical, data, and imaging coordinating centers and the NICHD. All information was collected by trained individuals using standardized protocols. Detailed information regarding study design and participant recruitment has been reported elsewhere [11, 12].

## Data collection and neonatal measurement

At enrollment, a baseline in-person interview was conducted to gather demographic characteristics and obstetric history, including self-reported maternal race/ethnicity (non-Hispanic White; non-Hispanic Black; Asian-Pacific Islander; Hispanic), maternal age, height (cm), pre-pregnancy weight (kg), education (<high school; high school; some college; college undergraduate; postgraduate college), and parity (0; 1; 2/+). Participants were followed through delivery with additional chart abstraction for infant sex (male; female) and birthweight (g). Based on birthweight, infants were categorized as low birthweight (<2500g), normal birthweight (2500-4000g), or macrosomic (>4000g) to determine clinical relevance of differences in birthweight.

After birth, trained research nurses performed an exam to obtain neonatal anthropometric measurements (time to exam: mean age 1.7 days, SD 3.5), as reported previously [12–15]. Non-skeletal measures included: mid-upper arm (MUAC), abdominal (AC; measured level midway between the xiphisternum and umbilicus), and mid-upper thigh circumferences (MUTC) measured with a non-stretch measuring tape placed directly over skin (cm); subscapular, triceps, abdominal flank, and anterior thigh skinfolds (mm) measured using a Lange skinfold caliper (Beta Technology, Inc., Santa Cruz, CA); and % fat mass [14, 16]. Skeletal measures (cm) included length (measured distance from soles of infants' feet to top of infants' heads in a supine position [Seca 416 infantometer; SECA, Hamburg, Germany]) and head circumference measured with a non-stretch measuring tape. Anthropometrics were measured in duplicate and averaged for analysis. If the second measure differed from the first by a pre-specified amount (expected technical error), a third measure was taken and the two closest measurements were averaged [13, 14].

## Blood collection and analysis

At enrollment, blood samples were collected (mean gestational age 12.7 weeks, SD 0.96), processed, and stored at −80˚C for banking following standardized protocol. Plasma samples were shipped on dry ice to the Wadsworth Center for quantification. Plasma nicotine and cotinine were measured using an ultra-performance liquid chromatography coupled with an electrospray triple quadrupole tandem mass spectrometry with limits of quantification (LOQ) of 0.05 ng/mL for cotinine and 0.13 ng/mL for nicotine. Samples were spiked with labeled internal standards of cotinine and nicotine and passed through Hybrid solid phase extraction cartridge (Phospholipid, 30 mg/1 mL, Supelco, Bellefonte, PA). The recoveries of cotinine and nicotine through the analytical method was 100%. Coefficients of variance were 11.6% for cotinine and 9.6% for nicotine. The standard reference material, SRM-3672, was analyzed to confirm the accuracy and precision of the method.

## Statistical analysis

Unless otherwise noted, consistent with contemporary practice, machine observed values of cotinine and nicotine were used for analyses [17, 18], including negative values for cotinine and nicotine resulting from the subtraction of the concentration of the blank from the measured concentration. In the main analysis, continuous plasma concentrations of cotinine and nicotine (ng/mL) were log-transformed (log[1+value]) then rescaled by their standard deviation (SD). Regression models were run on these scaled and log-transformed concentrations. Estimated regression coefficients and their corresponding 95% confidence intervals were then rescaled back for ease of interpretation in terms of 1-unit change in log-concentration, as presented in tables.

In secondary analyses, passive smoking exposure was also evaluated based on relevant bio-marker concentration cut-points to verify consistency across analytic techniques. While various cutpoints have been reported to distinguish non-smokers from passive/active smokers based on plasma cotinine concentration [3, 9, 19–21], we chose the cutpoint of 1 ng/mL (<1 ng/mL: unexposed/typical passive smoke exposure; ≥1 ng/mL cotinine: smoke exposure) [3, 21] to maximize sensitivity. Additionally, we evaluated passive smoke exposure based on plasma cotinine concentration above (exposed) or below the LOQ of both nicotine and cotinine (separately) [9, 19, 20]. Because of the relatively longer half-life of cotinine compared to nicotine [22] and because using biomarker categorizations (non-smokers versus passive/active smokers or above/below LOQ) resulted in sparse cells, we focused our main analysis on continuous cotinine concentration, with other exposure variables reported to verify consistency across biomarkers and biomarker categorizations.

Maternal and neonatal characteristics were described and compared by above/below LOQ-$_{cotinine}$ using 2-sided t-tests or chi-square tests to describe women with and without measured concentrations. Level of smoking exposure within our sample was determined overall (median, inter-quartile range [IQR]), by race/ethnicity, and by inclusion/exclusion in the current study. Differences were evaluated using Kruskal-Wallis nonparametric or chi-square tests (Fishers exact test used where cell <5).

For the main analysis based on continuous cotinine concentration, general linear regression (logistic regression for low birthweight and macrosomia in comparison to normal birthweight) was used to determine if the association between early gestation cotinine concentration and each neonatal anthropometric measure differed by race/ethnicity by including a cotinine x race/ethnicity interaction term in the models (p<0.1 for Type III SS considered statistically significant), first in unadjusted and then in adjusted multivariable models including confounders: maternal age, height, pre-pregnancy weight, education, parity, and infant sex (as defined above). Model-derived race/ethnicity specific estimates and 95% CIs of each cotinine-neonatal anthropometric measure association were generated, adjusting for time to exam (except for birthweight). We chose not to adjust for gestational age at birth because it is an intermediary in the association between the exposures and anthropometric outcomes and thus adjustment would introduce bias [23]. To confirm main analysis results, the above statistical methods were repeated for nicotine concentration and for cotinine and nicotine cut-points (smoke exposure vs non-smoker; above vs below LOQ; low birthweight and macrosomia not evaluated due to small number of events). Due to our interest in exploring potential differences by race/ethnicity and for consistency across models, all regression results are presented by race/ethnicity. We also fitted splines for assessing the functional form of the association between smoking biomarkers and anthropometric measurements.

Sensitivity analyses, using main analysis methods, were performed to determine if results were consistent when restricting the cohort to comprise only term births among women without gravid diseases or event (liveborn infant ≥37 weeks without pregnancy-related complications; without fetal anomalies) [11].

All analyses were conducted using SAS statistical software, with 2-sided tests using p<0.05 to determine statistical significance (unless otherwise specified).

## Results

Of 2334 participants in the Fetal Growth Studies-Singletons, 14 women were ineligible following enrollment, 186 did not have live births, 28 did not provide a baseline blood sample, 5 had insufficient sample for analysis, and 46 did not consent to the use of their blood sample, resulting in a study cohort comprising 2055 women for analysis. For analyses of neonatal

anthropometrics measured at the study exam, participants missing time to exam were excluded (n = 120); for analyses of skinfold measures, participants at 1 site which used incorrect calipers were also excluded (n = 129); for analyses of % fat mass, women with infants born at <37 weeks gestational age or with birthweight <2000 g (n = 99) or with negative or missing values of % fat mass were excluded as well (formula used not applicable for these neonates; n = 58; S1 Fig).

In this racially/ethnically diverse cohort (27% non-Hispanic White, 25% non-Hispanic Black, 28% Hispanic, 19% Asian-Pacific Islander), most women were educated (73% completed at least some college), married (76%), and had private health insurance/managed care (66%), while roughly half were nulliparous (49%). The secondary sex ratio was as expected with 52% of male infants (Table 1).

Women with cotinine concentration $\geq$LOQ$_{cotinine}$ were younger, less educated, more likely to be non-Hispanic Black, and to have a low birthweight infant, and less likely to be married, have private health insurance/managed care, and have a low-risk/uncomplicated pregnancy, and had a shorter time to study exam than women with cotinine <LOQ$_{cotinine}$ ($P$<0.05). No differences in cotinine or nicotine concentration were found between the included and excluded population (S1 Table).

Plasma concentrations of cotinine and nicotine were low, confirming our cohort was comprised largely of non-smokers (97%; Table 2). Our cohort had lower cotinine concentration than that of the nationally-representative sample of pregnant women reported by NHANES 2003–4, which found a median serum cotinine concentration of 0.03ng/mL and 66% of women with concentrations above LOQ$_{cotinine}$ (0.015ng/mL) [24]. Cotinine and nicotine concentration differed by race/ethnicity ($P$<0.001), and each was consistently higher in non-Hispanic Black versus other race women suggesting they may be an at risk population.

Of the 22% of women with a cotinine concentration above LOQ$_{cotinine}$, median (IQR) cotinine concentration was 0.17 ng/mL (0.08, 0.44). Similarly, of the 14% of women with a nicotine concentration above LOQ$_{nicotine}$, median (IQR) nicotine concentration was 0.25 ng/mL (0.20, 0.42). Considering these women with biomarker concentrations above the LOQ, cotinine and nicotine concentrations were highest among non-Hispanic Black women (S2 and S3 Figs). Neonatal anthropometric measures differed by race/ethnicity as previously reported [13].

Racial/ethnic differences were found in the association between cotinine concentration and birthweight ($P_{interaction}$ = 0.03 in adjusted model; S2 and S3 Tables; Fig 1A). In the adjusted model, for each 1 log-unit increase in cotinine concentration, birthweight increased 48 g (95% CI -44, 139) in non-Hispanic White women and 51 g (95%CI-81, 183) in Hispanic women, but decreased -90 g (95%CI -490, 309) in Asian-Pacific Islander and -93 g (95%CI -151, -35) in non-Hispanic Black women (Fig 1A). No racial/ethnic differences were found in the association between cotinine and skeletal measures, though among non-Hispanic Black women for each 1 log-unit increase in cotinine concentration head circumference decreased by -0.20 cm (95%CI −0.38, −0.02). Rather, racial/ethnic differences in associations between birthweight and cotinine concentration were seemingly driven by racial/ethnic differences in associations between cotinine and non-skeletal measures (AC, MUTC, and subscapular, triceps, and anterior thigh skinfolds; $P_{interaction}$<0.1; S2 Table) as evidenced by generally similar patterns of association as seen with birthweight within each racial/ethnic group (Fig 1A; S3 Table), though the patterns were less consistent among Hispanic women.

For nicotine concentration, similar racial/ethnic differences were noted for non-skeletal measures (S2 Table). However, examining the patterns within racial ethnic groups, while increasing nicotine concentrations were associated with increased size among non-Hispanic White women but decreasing size among non-Hispanic Black women, results were

**Table 1. Study cohort characteristics by categories of cotinine concentration, NICHD Fetal Growth Studies-Singletons 2009–2013 (n = 2055).**

|  | Overall N = 2055 | <LOQcotinine[a] (n = 1605) | ≥LOQcotinine[a] (n = 450) | p-value[b] |
|---|---|---|---|---|
| *Maternal characteristics* |  |  |  |  |
| Age (years; mean [SD]) | 28.2 (5.5) | 29.1 (5.2) | 25.2 (5.3) | < .001 |
| Race/ethnicity |  |  |  |  |
| Non-Hispanic White | 562 (27.3) | 506 (31.5) | 56 (12.4) | < .001 |
| Non-Hispanic Black | 518 (25.2) | 266 (16.6) | 252 (56.0) |  |
| Hispanic | 580 (28.2) | 495 (30.8) | 85 (18.9) |  |
| Asian/Pacific Islander | 395 (19.2) | 338 (21.1) | 57 (12.7) |  |
| Education |  |  |  |  |
| <High school | 207 (10.1) | 138 (8.6) | 69 (15.3) | < .001 |
| High school | 356 (17.3) | 218 (13.6) | 138 (30.7) |  |
| Some college | 598 (29.1) | 440 (27.4) | 158 (35.1) |  |
| College undergraduate | 515 (25.1) | 456 (28.4) | 59 (13.1) |  |
| Postgraduate college | 379 (18.4) | 353 (22.0) | 26 (5.8) |  |
| Marital status |  |  |  |  |
| Married/living as married | 1569 (76.4) | 1328 (82.8) | 241 (53.6) | < .001 |
| Not married | 484 (23.6) | 275 (17.2) | 209 (46.4) |  |
| Health insurance |  |  |  |  |
| Private/managed care | 1347 (65.5) | 1150 (71.7) | 197 (43.8) | < .001 |
| Other | 708 (34.5) | 455 (28.3) | 253 (56.2) |  |
| Parity |  |  |  |  |
| 0 | 1007 (49.0) | 775 (48.3) | 232 (51.6) | 0.052 |
| 1 | 703 (34.2) | 570 (35.5) | 133 (29.6) |  |
| 2/+ | 345 (16.8) | 260 (16.2) | 85 (18.9) |  |
| Maternal height (cm; mean [SD]) | 162 (6.9) | 162 (6.9) | 163 (7.0) | 0.050 |
| Maternal weight (kg; mean [SD]) | 62.4 (9.6) | 62.2 (9.4) | 63.2 (10.3) | 0.076 |
| Pre-pregnancy BMI (kg/m$^2$; mean [SD]) | 23.6 (3.1) | 23.6 (3.0) | 23.7 (3.4) | 0.403 |
| Low risk/uncomplicated[c] |  |  |  |  |
| Yes | 1676 (81.6) | 1327 (82.7) | 349 (77.6) | 0.013 |
| No | 379 (18.4) | 278 (17.3) | 101 (22.4) |  |
| *Neonatal characteristics* |  |  |  |  |
| Neonatal sex |  |  |  |  |
| Male | 1058 (51.7) | 844 (52.8) | 214 (47.7) | 0.052 |
| Female | 988 (48.3) | 753 (47.2) | 235 (52.3) |  |
| Birthweight (g; mean [SD]) | 3320 (500) | 3350 (486) | 3230 (536) | < .001 |
| Low birthweight (<2500 g) | 101 (4.9) | 64 (4.0) | 37 (8.3) | < .001 |
| Normal birthweight (2500–4000 g) | 1793 (87.8) | 1403 (88.1) | 390 (87.1) |  |
| Macrosomia (>4000 g) | 147 (7.2) | 126 (7.9) | 21 (4.7) |  |
| Preterm birth |  |  |  |  |
| No (≥37 weeks gestation) | 1925 (94.1) | 1508 (94.5) | 417 (92.9) | 0.199 |
| Yes (<37 weeks gestation) | 120 (5.9) | 88 (5.5) | 32 (7.1) |  |
| Time to exam (days; mean [SD])[d] | 1.73 (3.5) | 1.79 (3.8) | 1.51 (2.0) | 0.046 |

Abbreviations: NICHD, *Eunice Kennedy Shriver* National Institute of Child Health and Human Development; LOQ, limit of quantification; SD, standard deviation; BMI, body mass index.

Percentages may not add to 100% due to rounding.

Missing: marital status, n = 2; maternal height, n = 11; maternal weight, n = 4; maternal BMI, n = 15; neonatal sex, n = 9; low birth weight, n = 14; preterm, n = 10; time to exam, n = 120.

[a]LOQ$_{cotinine}$ = 0.05 ng/mL.

[b]Chi-squared for categorical variables; 2-sided t-test for continuous.

[c]Live-birth; term delivery ≥37 weeks; did not develop pregnancy-related complications; without fetal anomalies [11].

[d]Time to exam: variable represents number of days between birth and study examination at which neonatal anthropometrics were measured.

**Table 2. Plasma cotinine and nicotine concentration by race/ethnicity, NICHD Fetal Growth Studies-Singletons 2009–2013 (n = 2055).**

| Biomarker, classification | Cohort | Non-Hispanic White (n = 562) | Non-Hispanic Black (n = 518) | Hispanic (n = 580) | Asian/Pacific Islander (n = 395) | p-value[a] |
|---|---|---|---|---|---|---|
| **Cotinine (ng/mL)** | | | | | | |
| Median (IQR)[b] | 0.009 (0.0, 0.039) | 0.006 (0.00, 0.019) | 0.043 (0.007, 0.24) | 0.006 (0.0, 0.025) | 0.006 (0.0, 0.025) | < .001 |
| Non-smoker; n (%)[c] | 1993 (97.0) | 554 (98.6) | 472 (91.0) | 574 (99.0) | 393 (99.5) | < .001 |
| Passive smoker; n (%)[c] | 62 (3.0) | 8 (1.4) | 46 (8.9) | 6 (1.0) | 2 (0.50) | |
| %≥LOQ; n (%)[d] | 450 (21.9) | 56 (10.0) | 252 (48.7) | 85 (14.7) | 57 (14.4) | < .001 |
| **Nicotine (ng/mL)** | | | | | | |
| Median (IQR)[b] | -0.007 (-0.039, 0.041) | -0.016 (-0.045, 0.021) | 0.011 (-0.020, 0.073) | -0.010 (-0.047, 0.036) | -0.011 (-0.046, 0.025) | < .001 |
| %≥LOQ; n (%)[e] | 282 (13.7) | 86 (15.3) | 88 (17.0) | 78 (13.6) | 30 (7.6) | < .001 |

Abbreviations: NICHD, *Eunice Kennedy Shriver* National Institute of Child Health and Human Development; IQR, inter-quartile range; LOQ, limit of quantification. Percentages may not add to 100% due to rounding.

[a]Kruskal-Wallis nonparametric tests conducted to compare medians of chemical concentration across race/ethnic groups for continuous variables; Chi-square test (or Fishers exact test if cell <5) conducted for categorical variables.

[b]Median (IQR) values reported are machine derived values.

[c]Non-smoker: unexposed/typical passive smoke exposure: <1 ng/mL; Passive smoker: ≥1 ng/mL cotinine.

[d]$LOQ_{cotinine}$ = 0.05 ng/mL.

[e]$LOQ_{nicotine}$ = 0.13 ng/mL.

inconsistent among both Asian-Pacific Islander and Hispanic women (S3 Table; Fig 1B). No differences were seen for clinical outcomes (low birthweight or macrosomia) by race/ethnicity (S2 Table), though the direction of associations was in line with findings for birthweight with increasing biomarker concentrations decreasing the odds of low birthweight among non-Hispanic White women and increasing the odds of low birthweight among non-Hispanic Black

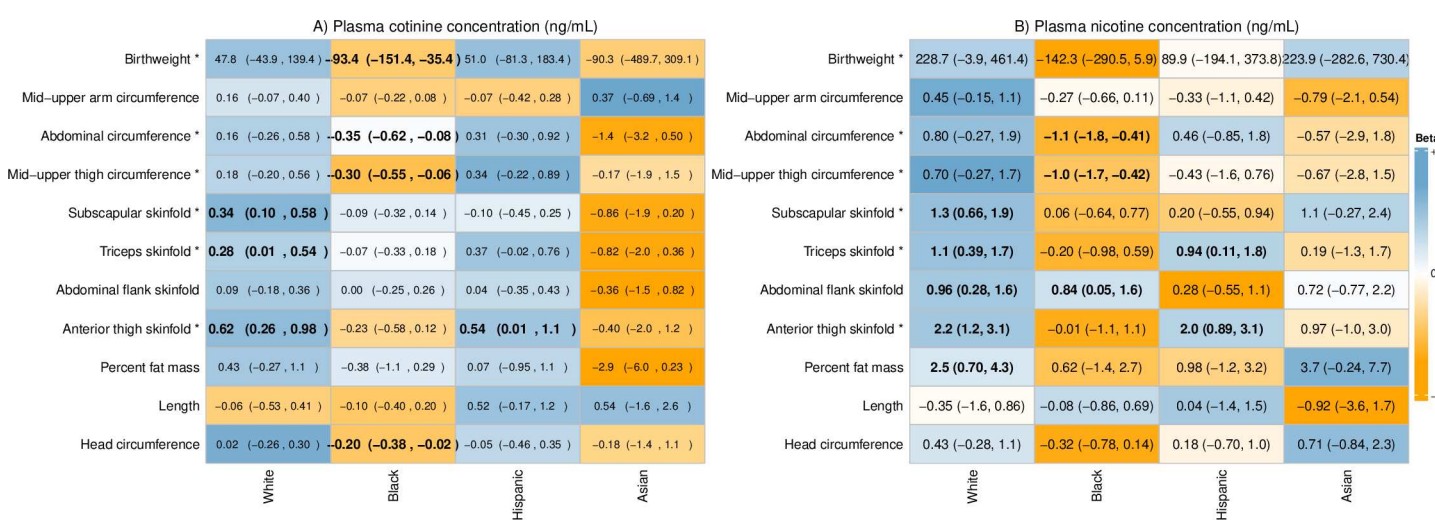

**Fig 1. Associations between plasma cotinine and nicotine concentrations and neonatal anthropometric measures by self-reported maternal race/ethnicity, NICHD Fetal Growth Studies-Singletons 2009–2013 (n = 2055).** Estimated association between plasma biomarker concentrations and neonatal anthropometric measures from adjusted multivariable generalized linear regression models, controlling for time to exam (except birthweight), infant sex, maternal age, height and weight, education, and parity. Results presented are the change in neonatal anthropometric measurements per 1-unit increase in log-transformed cotinine and nicotine plasma concentration and 95% confidence interval. For each neonatal anthropometric measure, the relative increase (blue) or decrease (orange) in size (relative to the standardized values of the beta) within each racial/ethnic group is demonstrated by the color gradient, with darker shades indicating stronger associations. *Statistically significant race/ethnicity x biomarker concentration interactions (p<0.1). BOLD: 95% confidence interval not crossing the null.

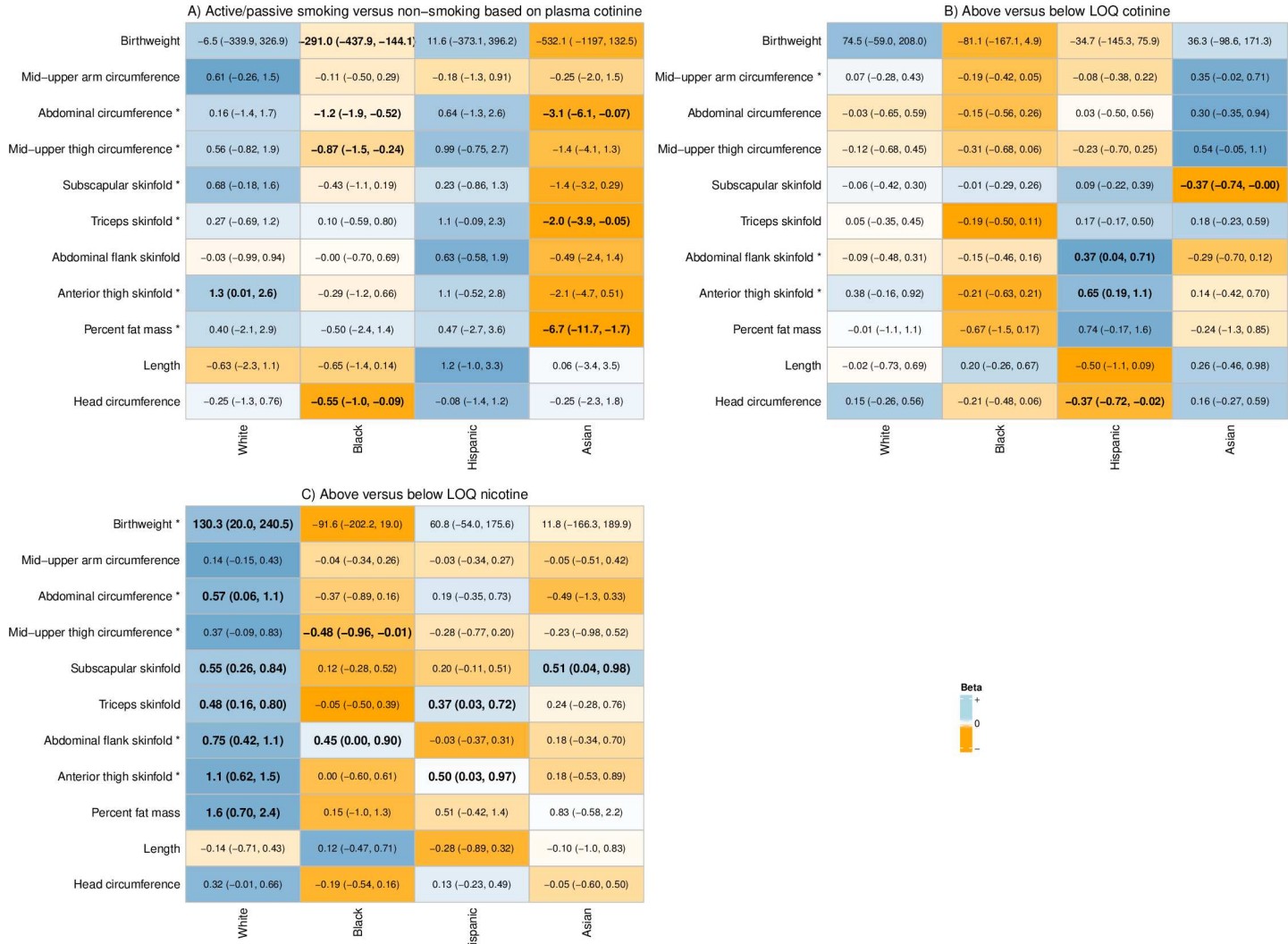

**Fig 2. Associations between relevant cotinine and nicotine concentration cut-points and neonatal anthropometric measures by self-reported maternal race/ethnicity, NICHD Fetal Growth Studies-Singletons 2009–2013 (n = 2055).** Estimated association between relevant biomarker cut-points (i.e. non-smoker versus passive smoker; above versus below limit of quantification [LOQ]) and neonatal anthropometric measures from adjusted multivariable generalized linear regression models, controlling for time to exam (except birthweight), infant sex, maternal age, height and weight, education, and parity. Results presented are the change in neonatal anthropometric measure among exposed relative to unexposed and 95% confidence interval. For each neonatal anthropometric measure, the relative increase (blue) or decrease (orange) in size (relative to the standardized values of the beta) within each racial/ethnic group is demonstrated by the color gradient, with darker shades indicating stronger associations. *Statistically significant race/ethnicity x biomarker concentration interactions (p<0.1). BOLD: 95% confidence interval not crossing the null.

women (S4 Table). No evidence of nonlinear associations was found (data from spline analysis not shown).

Consistent with results from the main analysis, racial/ethnic differences were found in associations between above/below $LOQ_{nicotine}$ and birthweight (p = 0.04; S2 Table). In general, similar opposing patterns were seen for the associations between cotinine and nicotine based on relevant exposure cut-points and non-skeletal measures comparing non-Hispanic Black and non-Hispanic White women, with decrements seen amongst non-Hispanic Black mothers (the group with the highest biomarker concentrations) and increases amongst non-Hispanic White mothers (S2 and S3 Tables; Fig 2). An inverse association between biomarker

concentration and head circumference among non-Hispanic Black women was also observed consistently (Figs 1 and 2; S3 Table).

## Sensitivity analysis

Plasma cotinine and nicotine concentrations did not differ between the main analysis cohort and the cohort of term births among women without complicated pregnancies reflecting gravid diseases (p>0.05; n = 1676). In this subgroup, racial/ethnic differences were not consistently observed in the associations between cotinine and nicotine concentration or their relevant cut-points and birthweight or other neonatal anthropometric measures (S5 Table). Nonetheless, similar opposing patterns as found in the main analysis were seen for the associations between cotinine and nicotine concentrations and non-skeletal measures by race/ethnicity, with decrements seen amongst non-Hispanic Black mothers and increases amongst non-Hispanic White mothers (S6 and S7 Tables). An inverse association between biomarker concentration and head circumference among non-Hispanic Black women was also consistently observed among women with term births and uncomplicated pregnancies.

## Discussion

In this cohort of pregnant women with low risk antenatal profiles carrying singleton pregnancies, overall, plasma nicotine and cotinine concentrations were low, but were higher among non-Hispanic Black women in comparison to non-Hispanic White, Hispanic, or Asian-Pacific Islander women. Though not all results were statistically significant, consistent trends were found, with cotinine concentration positively associated with birthweight and non-skeletal measures in non-Hispanic White women (and to a lesser extent among Hispanic women) but negatively in non-Hispanic Black women (and to a lesser extent among Asian-Pacific Islander women), potentially reflecting higher biomarker concentrations found in the latter subgroup. Though statistically significant differences in the associations between smoking exposure and clinical outcomes (low birthweight, macrosomia) by race/ethnicity were not observed, trends were similar for these outcomes comparing non-Hispanic White women and non-Hispanic Black women. Results were similar for nicotine and both biomarkers when categorized and in a sensitivity analysis. No differences in the association between biomarker concentrations and skeletal measures by race/ethnicity were observed, though decreases in head circumference with increasing biomarker exposure were found only among non-Hispanic Black women.

The low plasma biomarker concentrations observed in our cohort highlight the success of public health campaigns to reduce smoking and second hand smoke exposure in the United States [3, 9]. However, the expected negative impact of plasma biomarker concentration on neonatal anthropometric measures was found consistently only among non-Hispanic Black women, the racial/ethnic group in our cohort with the highest plasma biomarker concentrations and at greater risk of exposure to passive smoking [9], though we found some evidence of a similar trend among Asian-Pacific Islander women. Potential differences in the association between passive smoke exposure and neonatal anthropometrics is supported by previous studies which found a stronger association between smoking and lower birthweight in Black compared to White women [25], though another study found the opposite in term, but not preterm, deliveries [26]. Further, nicotine metabolism is reported to be slower and plasma biomarker levels higher at a given exposure level in both non-Hispanic Black and Asian women than in non-Hispanic White and Hispanic women [5, 10], due at least in part to genetic differences in nicotine and cotinine metabolism [27, 28]. Therefore, the negative association between biomarker concentration and neonatal anthropometrics among non-smoking non-Hispanic Black and Asian-Pacific Islander woman even at very low levels of exposure could be

the result of prolonged clearance time, particularly when coupled with the greater initial exposure among non-Hispanic Black women. Though genetic differences contribute to heterogeneity in nicotine metabolism, we did not find genetic differences in our cohort based on a genome-wide association study (GWAS) of single-nucleotide polymorphisms associated with nicotine metabolism in previous studies (data not shown). As they represent a high risk population, future public health campaigns should target non-Hispanic Black women to eliminate this health disparity. Additional research is needed to disentangle underlying biologic/genetic versus socio-economic factors.

In contrast, our finding of increases in non-skeletal anthropometric measures with increasing cotinine or nicotine concentration among non-Hispanic White infants, and to some extent among Hispanic women, was unexpected. Given the importance of socio-economic disadvantage to smoking [1], and to fetal growth [29], for non-Hispanic White women with healthy pregnancies, limited negative effects of very low levels of passive smoking exposure on fetal growth would be plausible. Factors underlying the Hispanic Paradox [30], which notes less low birth weight among Hispanic women, could similarly explain limited negative effects of passive smoking on neonatal anthropometrics in that group. In relation to nicotine metabolism, the quicker clearance in these groups compared to non-Hispanic Black and Asian women could also prevent decreases neonatal anthropometric measures at very low levels of exposure among otherwise healthy women. Interestingly, though results were not significant and were part of post-hoc analyses, a previous study in a majority White population of early pregnancy smokers who subsequently continued smoking, quit, or partially quit found that having 1, 2, or 4 smokers in the home decreased birthweight and length compared to homes without smokers, while having 3 smokers in the home increased birthweight overall and in term births [1], supporting our finding that in some cases passive smoke exposure among White women may be associated with increased infant size. Nonetheless, the potential mechanisms leading to positive association between passive smoke exposure and neonatal size in these groups is unclear and warrants additional research.

Our finding of an association between passive smoking and birthweight and non-skeletal measures specifically is supported by studies in mice, which found that nicotine exposure reduced abdominal and visceral fat [31] and perinatal exposure increased body weight and subcutaneous and visceral fat mass later in life [32]. Overall, studies in humans are consistent with our findings related to the association between smoking exposure and non-skeletal (exposure associated with decreased birthweight, fat measures, and abdominal circumference [33, 34], though no differences in skinfolds [34, 35] or limb circumferences [35]). However, previous studies have also found associations between smoking exposure and skeletal measures (exposure associated with having reduced infant/limb length [33–35], head circumference [34], biparietal diameter [33], and fat free mass [35]). Differences between studies may be due to differences in exposure assessment (self-report versus biomarkers) and categorizations, study population (active smokers included or excluded), and temporal changes in exposure level.

Strengths of this study include the inclusion of a racially/ethnically diverse population of pregnant women from across the US and standardized neonatal anthropometric assessment. By focusing on a population of self-reported non-smokers, we examined low level of smoking exposure and potential implications for neonatal anthropometry. Because cohort inclusion criteria relied on self-reported smoking status which is subject to response bias, our study population may have included some smokers, though the low biomarker concentrations observed suggest our study was largely comprised of non-smokers. Because our study is observational in nature and additional variables may be associated with passive smoking and neonatal size (for example, lifestyle factors and gestational weight gain) and may vary by race/ethnicity, residual

confounding is possible. However, in additional analyses we found that physical activity (based on metabolic equivalent of task hours per week) was not associated with plasma cotinine concentration above the LOQ and that the inclusion of nutrition variables (nutrition factors captured with the Alternative Healthy Eating Index [36] and total caloric intake derived from the Food Frequency Questionnaire) in adjusted analyses did not alter our results (data not shown). Though we were only able to assess biomarker concentrations at one time-point and measurement error in our biomarker assessments is possible, the consistent results across exposure and outcome measures provides a robust assessment of early pregnancy passive smoking exposure and suggests additional comprehensive, longitudinal research is needed to assess exposure at different time points in pregnancy.

## Conclusions

In this diverse longitudinal cohort comprising non-smoking pregnant women with low risk antenatal profiles, we found that passive smoking exposure as measured by plasma nicotine and cotinine concentration was associated with decrements in neonatal size for non-Hispanic Black women, the group with the highest plasma biomarker concentrations. Collectively, our findings underscore the beneficial impact of public health initiatives aimed at reducing smoking and its associated exposure for pregnant women and fetuses. Still, targeted interventions for further reduction in exposure may be warranted for non-Hispanic Black women.

## Supporting information

**S1 Fig. Participant flowchart.**
(DOCX)

**S2 Fig. Distribution of plasma cotinine concentrations above the LOQ overall and by race/ethnicity, NICHD Fetal Growth Studies-Singletons 2009–2013.**
(DOCX)

**S3 Fig. Distribution of plasma nicotine concentrations above the LOQ overall and by race/ethnicity, NICHD Fetal Growth Studies-Singletons 2009–2013.**
(DOCX)

**S1 Table. Comparison of plasma biomarker concentration among included versus excluded women.**
(DOCX)

**S2 Table. Interaction by race/ethnicity in the plasma biomarker concentrations-neonatal anthropometrics association among non-smoking pregnant women.**
(DOCX)

**S3 Table. Plasma biomarker concentration-neonatal anthropometrics associations by race/ethnicity among non-smoking pregnant women (unadjusted models).**
(DOCX)

**S4 Table. Plasma biomarker concentration-clinical outcomes associations by race/ethnicity among non-smoking pregnant women.**
(DOCX)

**S5 Table. Interaction by race/ethnicity in plasma biomarker concentrations-neonatal anthropometrics association among non-smoking pregnant women (sensitivity analyses in the standard population).**
(DOCX)

**S6 Table. Continuous plasma biomarker concentration-neonatal anthropometrics associations by race/ethnicity in standard population of non-smoking pregnant women.**
(DOCX)

**S7 Table. Plasma biomarker concentration cut-points-neonatal anthropometrics associations by race/ethnicity in standard population of non-smoking pregnant women.**
(DOCX)

## Acknowledgments

We acknowledge the effort of research teams at all participating clinical centers, i.e., Christina Care Health Systems, Columbia University, Fountain Valley Hospital, California, Long Beach Memorial Medical Center, New York Hospital, Queens, Northwestern University, University of Alabama at Birmingham, University of California, Irvine, Medical University of South Carolina, Saint Peters University Hospital, Tufts University, and Women and Infants Hospital of Rhode Island. The authors also acknowledge the Wadsworth Center, C-TASC, and The EMMES Corporations in providing data and imaging support for this multi-site study.

## Author Contributions

**Conceptualization:** Melissa M. Amyx, Rajeshwari Sundaram, Germaine M. Buck Louis, Nicole M. Gerlanc, Alaina M. Bever, Kurunthachalam Kannan, Melissa M. Smarr, Fasil Tekola-Ayele, Cuilin Zhang, Katherine L. Grantz.

**Data curation:** Kurunthachalam Kannan, Morgan Robinson, Dian He, Katherine L. Grantz.

**Formal analysis:** Melissa M. Amyx, Rajeshwari Sundaram, Nicole M. Gerlanc, Dian He.

**Investigation:** Cuilin Zhang, Katherine L. Grantz.

**Methodology:** Rajeshwari Sundaram, Dian He, Fasil Tekola-Ayele.

**Project administration:** Germaine M. Buck Louis, Cuilin Zhang, Katherine L. Grantz.

**Supervision:** Rajeshwari Sundaram, Katherine L. Grantz.

**Writing – original draft:** Melissa M. Amyx.

**Writing – review & editing:** Rajeshwari Sundaram, Germaine M. Buck Louis, Nicole M. Gerlanc, Alaina M. Bever, Kurunthachalam Kannan, Morgan Robinson, Melissa M. Smarr, Dian He, Fasil Tekola-Ayele, Cuilin Zhang, Katherine L. Grantz.

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
