## [Decision Letter · Decision Letter 0]

31 May 2021

PONE-D-21-14641

Association between early gestation passive smoke exposure and neonatal size among self-reported non-smoking women by race/ethnicity: a cohort study

PLOS ONE

Dear Dr. Grantz,

Thank you for submitting your manuscript to PLOS ONE. After careful consideration, we feel that it has merit but does not fully meet PLOS ONE’s publication criteria as it currently stands. Therefore, we invite you to submit a revised version of the manuscript that addresses the points raised during the review process.

This study is well-written and interesting.  However, I do agree with the reviewers' request for more details in the methods section (i.e., about the biomarkers).  Similar to Reviewer 2, I also found the figures difficult to follow and, additionally, am concerned about the color scheme being sensitive to color-blind readers.

We look forward to receiving your revised manuscript.

Kind regards,

Sze Yan Liu, PhD

Academic Editor

PLOS ONE

Journal Requirements:

We note that one or more of the authors are employed by a commercial company: The Prospective Group, Inc.

3.1. Please provide an amended Funding Statement declaring this commercial affiliation, as well as a statement regarding the Role of Funders in your study. If the funding organization did not play a role in the study design, data collection and analysis, decision to publish, or preparation of the manuscript and only provided financial support in the form of authors' salaries and/or research materials, please review your statements relating to the author contributions, and ensure you have specifically and accurately indicated the role(s) that these authors had in your study. You can update author roles in the Author Contributions section of the online submission form.

3.2. Please also provide an updated Competing Interests Statement declaring this commercial affiliation along with any other relevant declarations relating to employment, consultancy, patents, products in development, or marketed products, etc.  

Reviewers' comments:

Reviewer's Responses to Questions

**Comments to the Author**

1. Is the manuscript technically sound, and do the data support the conclusions?

Reviewer #1: Yes

Reviewer #2: Yes

2. Has the statistical analysis been performed appropriately and rigorously? 

Reviewer #1: Yes

Reviewer #2: Yes

3. Have the authors made all data underlying the findings in their manuscript fully available?

Reviewer #1: Yes

Reviewer #2: Yes

4. Is the manuscript presented in an intelligible fashion and written in standard English?

Reviewer #1: Yes

Reviewer #2: Yes

5. Review Comments to the Author

Reviewer #1: * in your abstract, you mentioned that you measures weight, 10 non-skeletal and 2-skeletal anthropometric measurements, but in the main manuscript (table 1) and supplemental data (S2 &S3), I could only find 8-non skeletal anthropometric measures (MUAC, C, MUHC, AC, TC, AFC, ATC, % Fat mass). Could you explain?

*in the methods section, how did you determine the cut off points for cotinine and nicotine? Could you explain explicitly in the methods section?

* in table 1, the age was significantly differed between both group, but i did not see any adjustment in the results section. Why? the same question also apply to marital status and health insurance.

* in the first paragraph of discussion you mentioned that "cotinine was positively associated with BW and all of non skeletal measures (...) in non hispanic white women......". But this result only consistent for Subscapular, triceps, and abdominal flank circumference, the other measures were not consistent because the 95% CIs included null value. I would suggest to revised this statement because this statement can be misleading.

Reviewer #2: This study aimed to examine the interaction between maternal exposure to secondhand smoke during pregnancy and maternal race/ethnicity on neonatal anthropometric measures. The authors observed that the associations between cotinine concentration and infant weight differed by race/ethnicity- where offspring born to exposed (as opposed to unexposed) White or Hispanic women were larger at birth whereas offspring born to exposed (as opposed to unexposed) Black or Asian women were smaller at birth. Other findings from the study consistently showed systematically smaller offspring born to exposed (as opposed to unexposed) Black or Asian women. The strengths of this study include the large sample size, the use of an ethnically diverse population, and the use of biomarkers to assess exposure to secondhand smoke. found that the secondhand smoke-birth weight association varies by race/ethnicity. Overall, this was a nicely written paper and the findings are very interesting. I have a few comments/questions for the authors as follows:

Consistency in capitalization of Black and White participants. (The authors switch to lowercase in the discussion).

Abstract, lines 12-15: The reference group here is unexposed in the same racial/ethnic group, correct? Please clarify.

Introduction, line 26: What is viz.?

Line 36: The authors might want to clarify that nicotine and cotinine are biomarkers of active AND passive smoking.

Lines 48-29: “This cohort presumably allows for the assessment of passive cigarette smoking exposure in relation to neonatal anthropometry.” I am not sure what this statement means.

Lines 114-117 “Because it is a more stable biomarker and small cell size/low power for analyses using biomarker categorizations, we focused our main analysis on cotinine concentration, with other exposure variables reported to verify consistency across biomarkers and biomarker categorizations.” A few suggestions: 1) include a citation for the improved “stability” of cotinine versus nicotine and 2) clarify what you mean by “small cell size/low power for analyses” (do the authors mean that there was less sparse cells for cotinine?)

Methods: Did the authors consider adjusting for gestational age at birth or restricting the analyses to term births? Some published studies have restricted to term delivery by analyses or by design. (e.g. https://www.nature.com/articles/srep24987). The results may not change, but this stuck out to me as an important consideration.

Table 1: I do believe the rows are flipped for preterm births.

Figures: Would tables be a better way to present the information? The same information presented in tables with bolded numbers (to denote 95% confidence interval not crossing the null) might be easier to read and easier on the eyes. This is my opinion though so I will defer to the editor.

Discussion:

The authors seem to focus on the finding that secondhand smoke is associated with smaller infant size among offspring born to exposed (as compared to unexposed) non-Hispanic Black and Asian women. This is consistent with the literature, which suggests that secondhand/passive smoking in pregnancy is associated with either no change or a slight decrease in birth weight (e.g. https://pubmed.ncbi.nlm.nih.gov/3752056/). So, the finding that infant size is larger among offspring born to exposed (as compared to unexposed) White and Hispanic women is quite surprising. The authors could briefly discuss why this might be, as I think this is a tremendously novel finding from this study.

The authors speculate about the mechanisms for the smaller infant size observed in offspring of Black women, but not Asian women. It would be useful to briefly discuss the underlying mechanisms for smaller infant size among Asian women. In particular, percent fat mass appears to be a lot lower. This may be particularly important because percent fat mass may be a surrogate for maternal exposures in pregnancy as well as childhood obesity. https://pubmed.ncbi.nlm.nih.gov/32796097/

Limitations: There are many potential confounders that are not measured/adjusted for that could have a huge impact on infant size at birth. In particular, gestational weight gain and maternal diet are very closely related to the exposure and outcome, and vary considerably within racial/ethnic groups. I think the authors should think carefully about how these variables could impact their analyses and perhaps soften some of the statements in the discussion. I do wonder if some of these racial/ethnic differences would weaken or disappear if the analyses had adjusted for these potential confounders.

6. PLOS authors have the option to publish the peer review history of their article (what does this mean?). If published, this will include your full peer review and any attached files.

Reviewer #1: **Yes: **Frida Soesanti

Reviewer #2: No

---

## [Author Response · Author response to Decision Letter 0]

29 Jul 2021

Thank you for the opportunity to revise our manuscript entitled “Association between early gestation passive smoke exposure and neonatal size among self-reported non-smoking women by race/ethnicity: a cohort study” for PLOS ONE. We found the reviews helpful and have revised our manuscript accordingly. We believe the manuscript has been significantly improved based on the helpful comments of the reviewers. Detailed point-by-point responses to the comments have been uploaded and are detailed below.

Reviewer #1:

* in your abstract, you mentioned that you measures weight, 10 non-skeletal and 2-skeletal anthropometric measurements, but in the main manuscript (table 1) and supplemental data (S2 &S3), I could only find 8-non skeletal anthropometric measures (MUAC, C, MUHC, AC, TC, AFC, ATC, % Fat mass). Could you explain?

Response: Thank you for your careful review and noting that an error was made when reporting the number of measures in the abstract. This has been rectified (page 3, lines 43-4): “Neonatal anthropometric measures included weight, 8 non-skeletal, and 2 skeletal measures.”

*in the methods section, how did you determine the cut off points for cotinine and nicotine? Could you explain explicitly in the methods section?

Response: This section was updated to clarify the selection of the cutpoints (page 8, line 150-63): “In secondary analyses, passive smoking exposure was also evaluated based on relevant biomarker concentration cut-points to verify consistency across analytic techniques. While various cutpoints have been reported to distinguish non-smokers from passive/active smokers based on plasma cotinine concentration,[3,9,19-21] we chose the cutpoint of 1 ng/mL (<1 ng/mL: unexposed/typical passive smoke exposure; ≥1 ng/mL cotinine: smoke exposure)[3,21] to maximize sensitivity. Additionally, we evaluated passive smoke exposure based on plasma cotinine concentration above (exposed) or below the LOQ of both nicotine and cotinine (separately). Because of the relatively longer half-life of cotinine compared to nicotine[22] and because using biomarker categorizations (non-smokers versus passive/active smokers or above/below LOQ) resulted in sparse cells, we focused our main analysis on continuous cotinine concentration, with other exposure variables reported to verify consistency across biomarkers and biomarker categorizations.”

* in table 1, the age was significantly differed between both group, but i did not see any adjustment in the results section. Why? the same question also apply to marital status and health insurance.

Response: In response to this request we have now updated all relevant analyses to include maternal age (eTables 2, 4, 5, 6, 7; throughout Abstract and Results sections). Results were consistent with our previous findings after adding this variable, and our conclusions were unchanged.

We also ran the analysis with marital status and health insurance (in addition to maternal age) included in the adjusted models and the results were consistent with those reported. However, for our final manuscript, we chose not to include these 2 variables as they are proxies for socioeconomic status and we had already included maternal education in the models, thus we were concerned that including these variables as well would result in overadjustment. 

* in the first paragraph of discussion you mentioned that "cotinine was positively associated with BW and all of non skeletal measures (...) in non hispanic white women......". But this result only consistent for Subscapular, triceps, and abdominal flank circumference, the other measures were not consistent because the 95% CIs included null value. I would suggest to revised this statement because this statement can be misleading.

Response: We have revised the first paragraph of the discussion to more accurately reflect our results as (page 19, lines 350-58): “Though not all results were statistically significant, consistent trends were found, with cotinine concentration positively associated with birthweight and non-skeletal measures in non-Hispanic White women (and to a lesser extent Hispanic women) but negatively in non-Hispanic Black women (and to a lesser extent among Asian-Pacific Islander women), potentially reflecting higher biomarker concentrations found in the latter subgroup.”

Reviewer #2: 

This study aimed to examine the interaction between maternal exposure to secondhand smoke during pregnancy and maternal race/ethnicity on neonatal anthropometric measures. The authors observed that the associations between cotinine concentration and infant weight differed by race/ethnicity- where offspring born to exposed (as opposed to unexposed) White or Hispanic women were larger at birth whereas offspring born to exposed (as opposed to unexposed) Black or Asian women were smaller at birth. Other findings from the study consistently showed systematically smaller offspring born to exposed (as opposed to unexposed) Black or Asian women. The strengths of this study include the large sample size, the use of an ethnically diverse population, and the use of biomarkers to assess exposure to secondhand smoke. found that the secondhand smoke-birth weight association varies by race/ethnicity. Overall, this was a nicely written paper and the findings are very interesting. I have a few comments/questions for the authors as follows:

Consistency in capitalization of Black and White participants. (The authors switch to lowercase in the discussion).

Response: The manuscript has been revised throughout to consistently capitalize Black and White.

Abstract, lines 12-15: The reference group here is unexposed in the same racial/ethnic group, correct? Please clarify.

Response: This sentence has been revised to clarify the reference group as (page 3, lines 47-52): “The association between cotinine concentration and infant weight differed by race/ethnicity (Pinteraction=0.034); compared to women of the same race/ethnicity, per 1 log-unit increase in cotinine, weight increased 48g (95%CI -44, 139) in White and 51g (95%CI -81, 183) in Hispanic women, but decreased -90g (95%CI -490, 309) in Asian and -93g (95%CI -151, -35) in Black women.”

Introduction, line 26: What is viz.?

Response: This sentence has been revised for clarity as (page 4, line 63-65): “Previous research has established that cigarette smoking negatively affects fetal growth[1,2] and birth size (for example, birthweight is reduced approximately 150-300g among women continuing to smoke during pregnancy).”

Line 36: The authors might want to clarify that nicotine and cotinine are biomarkers of active AND passive smoking.

Response: This sentence has been revised for clarity as (page 4, line 71-73): “Nicotine, and its metabolite cotinine, serves as a biomarker for measuring tobacco smoking exposure, both active and passive, and can readily cross the placenta.[7,8] Further, nicotine has been directly implicated as having deleterious effects on fetal growth.[8]”

Lines 48-29: “This cohort presumably allows for the assessment of passive cigarette smoking exposure in relation to neonatal anthropometry.” I am not sure what this statement means.

Response: This sentence has been revised for clarity as (page 5, lines 84-9): “Therefore, our objective was to determine if the relationship of plasma concentrations of nicotine and cotinine with neonatal anthropometry differed by race/ethnicity among nonsmoking pregnant women (whose biomarker levels would indicate passive smoke exposure) with low-risk antenatal profiles.”

Lines 114-117 “Because it is a more stable biomarker and small cell size/low power for analyses using biomarker categorizations, we focused our main analysis on cotinine concentration, with other exposure variables reported to verify consistency across biomarkers and biomarker categorizations.” A few suggestions: 1) include a citation for the improved “stability” of cotinine versus nicotine and 2) clarify what you mean by “small cell size/low power for analyses” (do the authors mean that there was less sparse cells for cotinine?)

Response: This sentence has been revised for clarity and a reference added (page 8, line 158-3): “Because of the relatively longer half-life of cotinine compared to nicotine[22] and because using biomarker categorizations (non-smokers versus passive/active smokers or above/below LOQ) resulted in sparse cells, we focused our main analysis on continuous cotinine concentration, with other exposure variables reported to verify consistency across biomarkers and biomarker categorizations.”

Methods: Did the authors consider adjusting for gestational age at birth or restricting the analyses to term births? Some published studies have restricted to term delivery by analyses or by design. (e.g. https://www.nature.com/articles/srep24987). The results may not change, but this stuck out to me as an important consideration.

Response: As mentioned in the Methods (page 9, lines 179-81), we “chose not to adjust for gestational age at birth because it is an intermediary in the association between the exposures and anthropometric outcomes and thus adjustment would introduce bias.” Additionally, the subpopulation included in our sensitivity analysis was restricted to women with term deliveries without pregnancy-related complications or fetal anomalies. We have revised relevant sections to highlight that this subpopulation was term deliveries only:

Page 9, lines 188-91: “Sensitivity analyses, using main analysis methods, were performed to determine if results were consistent when restricting the cohort to comprise only term births among women without gravid diseases or event (liveborn infant ≥37 weeks without pregnancy-related complications; without fetal anomalies)”

Page 18, lines 333-5: “Plasma cotinine and nicotine concentrations did not differ between the main analysis cohort and the cohort of term births among women without complicated pregnancies reflecting gravid diseases”

Page 19, lines 342-4: “An inverse association between biomarker concentration and head circumference among non-Hispanic black women was also consistently observed among women with term births and uncomplicated pregnancies.”

Table 1: I do believe the rows are flipped for preterm births.

Response: Thank you for pointing out this error, which we have corrected.

Figures: Would tables be a better way to present the information? The same information presented in tables with bolded numbers (to denote 95% confidence interval not crossing the null) might be easier to read and easier on the eyes. This is my opinion though so I will defer to the editor.

Response: Given the amount of data in the figure and the number of tables already included in the manuscript, we feel that this data is best presented in a figure. To improve the clarity of the information presented in the figure, we have updated the figure legends. 

Page 16, lines 280-90: “Estimated association between plasma biomarker concentrations and neonatal anthropometric measures from adjusted multivariable generalized linear regression models, controlling for time to exam (except birthweight), infant sex, maternal age, height and weight, education, and parity. Results presented are the change in neonatal anthropometric measurements per 1-unit increase in log-transformed cotinine and nicotine plasma concentration and 95% confidence interval. For each neonatal anthropometric measure, the relative increase (blue) or decrease (orange) in size (relative to the standardized values of the beta) within each racial/ethnic group is demonstrated by the color gradient, with darker shades indicating stronger associations.”

Page 18, line 318-28: “Estimated association between relevant biomarker cut-points (i.e. non-smoker versus passive smoker; above versus below limit of quantification) and neonatal anthropometric measures from adjusted multivariable generalized linear regression models, controlling for time to exam (except birthweight), infant sex, maternal age, height and weight, education, and parity. Results presented are the change in neonatal anthropometric measure among exposed relative to unexposed and 95% confidence interval. For each neonatal anthropometric measure, the relative increase (blue) or decrease (orange) in size (relative to the standardized values of the beta) within each racial/ethnic group is demonstrated by the color gradient, with darker shades indicating stronger associations.”

As suggested by the Editor, we changed to color scheme of the figures to be color-blind friendly (blue and orange).

Discussion:

The authors seem to focus on the finding that secondhand smoke is associated with smaller infant size among offspring born to exposed (as compared to unexposed) non-Hispanic Black and Asian women. This is consistent with the literature, which suggests that secondhand/passive smoking in pregnancy is associated with either no change or a slight decrease in birth weight (e.g. https://pubmed.ncbi.nlm.nih.gov/3752056/). So, the finding that infant size is larger among offspring born to exposed (as compared to unexposed) White and Hispanic women is quite surprising. The authors could briefly discuss why this might be, as I think this is a tremendously novel finding from this study.

The authors speculate about the mechanisms for the smaller infant size observed in offspring of Black women, but not Asian women. It would be useful to briefly discuss the underlying mechanisms for smaller infant size among Asian women. In particular, percent fat mass appears to be a lot lower. This may be particularly important because percent fat mass may be a surrogate for maternal exposures in pregnancy as well as childhood obesity. https://pubmed.ncbi.nlm.nih.gov/32796097/

Response: Thank you for these suggestions. We have revised our Discussion extensively to highlight this novel finding and the potential mechanisms among Asian women.

Page 20, line 368-page 22, line 414: “However, the expected negative impact of plasma biomarker concentration on neonatal anthropometric measures was found consistently only among non-Hispanic Black women, the racial/ethnic group in our cohort with the highest plasma biomarker concentrations and at greater risk of exposure to passive smoking,[9] though we found some evidence of a similar trend among Asian-Pacific Islander women. Potential differences in the association between passive smoke exposure and neonatal anthropometrics is supported by previous studies which found a stronger association between smoking and lower birthweight in Black compared to White women,[25] though another study found the opposite in term, but not preterm, deliveries.[26] Further, nicotine metabolism is reported to be slower in both non-Hispanic Black and Asian women and plasma biomarker levels higher at a given exposure level than non-Hispanic White and Hispanic women,[5,10] due at least in part to genetic differences in nicotine and cotinine metabolism.[27,28] Therefore, the negative association between biomarker concentration and neonatal anthropometrics among non-smoking non-Hispanic Black and Asian-Pacific Islander woman even at very low levels of exposure could be the result of prolonged clearance time, particularly when coupled with the greater initial exposure among non-Hispanic Black women. Though genetic differences contribute to heterogeneity in nicotine metabolism, we did not find genetic differences in our cohort based on a genome-wide association study (GWAS) of single-nucleotide polymorphisms associated with nicotine metabolism in previous studies (data not shown). As they represent a high risk population, future public health campaigns should target non-Hispanic Black women to eliminate this health disparity. Additional research is needed to disentangle underlying biologic/genetic versus socio-economic factors.

In contrast, our finding of increases in non-skeletal anthropometric measures with increasing cotinine or nicotine concentration among non-Hispanic White infants, and to some extent among Hispanic women, was unexpected. Given the importance of socio-economic disadvantage to smoking[1] and to fetal growth,[29] for non-Hispanic White women with healthy pregnancies, limited negative effects of very low levels of passive smoking exposure on fetal growth would be plausible. Factors underlying the Hispanic Paradox,[30] which notes less low birth weight among Hispanic women, could similarly explain limited negative effects of passive smoking on neonatal anthropometrics in that group. In relation to nicotine metabolism, the quicker clearance in these groups compared to non-Hispanic Black and Asian women could also prevent decreases neonatal anthropometric measures at very low levels of exposure among otherwise healthy women. Interestingly, though results were not significant and were part of post-hoc analyses, a previous study in a majority White population of early pregnancy smokers who subsequently continued smoking, quit, or partially quit found that having 1, 2, or 4 smokers in the home decreased birthweight and length compared to homes without smokers, while having 3 smokers in the home increased birthweight overall and in term births,[1] supporting our finding that in some cases passive smoke exposure among White women may be associated with increased infant size. Nonetheless, the potential mechanisms leading to positive association between passive smoke exposure and neonatal size in these groups is unclear and warrants additional research.

Our finding of an association between passive smoking and birthweight and non-skeletal measures specifically is supported by studies in mice, which found that nicotine exposure reduced abdominal and visceral fat[31] and perinatal exposure increased body weight and subcutaneous and visceral fat mass later in life.[32]”

Limitations: There are many potential confounders that are not measured/adjusted for that could have a huge impact on infant size at birth. In particular, gestational weight gain and maternal diet are very closely related to the exposure and outcome, and vary considerably within racial/ethnic groups. I think the authors should think carefully about how these variables could impact their analyses and perhaps soften some of the statements in the discussion. I do wonder if some of these racial/ethnic differences would weaken or disappear if the analyses had adjusted for these potential confounders.

Response: In analyses not included in this manuscript, we found that the inclusion of nutrition variables did not change our results and that physical activity was not associated with passive smoking exposure (data not shown). We have included a statement in discussion to highlight the potential for residual confounding and note this analysis (page 24, line 455-66): “Because our study is observational in nature and additional variables may be associated with passive smoking and neonatal size (for example, lifestyle factors and gestational weight gain) and may vary by race/ethnicity, residual confounding is possible. However, we found in additional analyses that physical activity (based on metabolic equivalent of task hours per week) was not associated with plasma cotinine concentration above the LOQ and that the inclusion of nutrition variables (nutrition factors captured with the Alternative Healthy Eating Index[36] and total caloric intake derived from the Food Frequency Questionnaire) in adjusted analyses did not alter our results (data not shown).”

---

## [Decision Letter · Decision Letter 1]

13 Aug 2021

Association between early gestation passive smoke exposure and neonatal size among self-reported non-smoking women by race/ethnicity: a cohort study

PONE-D-21-14641R1

Dear Dr. Grantz,

We’re pleased to inform you that your manuscript has been judged scientifically suitable for publication and will be formally accepted for publication once it meets all outstanding technical requirements.

Kind regards,

Sze Yan Liu, PhD

Academic Editor

PLOS ONE

Additional Editor Comments (optional):

The authors did an excellent job addressing concerns from earlier submissions. 

Reviewers' comments:

Reviewer's Responses to Questions

**Comments to the Author**

1. If the authors have adequately addressed your comments raised in a previous round of review and you feel that this manuscript is now acceptable for publication, you may indicate that here to bypass the “Comments to the Author” section, enter your conflict of interest statement in the “Confidential to Editor” section, and submit your "Accept" recommendation.

Reviewer #1: All comments have been addressed

Reviewer #2: All comments have been addressed

2. Is the manuscript technically sound, and do the data support the conclusions?

Reviewer #1: Yes

Reviewer #2: Yes

3. Has the statistical analysis been performed appropriately and rigorously? 

Reviewer #1: Yes

Reviewer #2: Yes

4. Have the authors made all data underlying the findings in their manuscript fully available?

Reviewer #1: Yes

Reviewer #2: Yes

5. Is the manuscript presented in an intelligible fashion and written in standard English?

Reviewer #1: Yes

Reviewer #2: Yes

6. Review Comments to the Author

Reviewer #1: All of my concerns have been well addressed by the authors. This manuscript is very well written and the conclusion was drawn based on robust data and appropriate statistical analysis.

Reviewer #2: Thank you for the opportunity to review the revised submission. I am impressed with the author's careful responses and attention to detail in their statistical analysis. Additionally, the discussion was very interesting to read. I appreciated that the authors outlined some very plausible potential mechanisms to explain the somewhat surprising result that increased cotinine concentrations were associated with increases in anthropometric measures in non-Hispanic White (and Hispanic women, to a lesser extent).

Overall, this was a pleasure to read. I believe this manuscript is ready for publication.

7. PLOS authors have the option to publish the peer review history of their article (what does this mean?). If published, this will include your full peer review and any attached files.

Reviewer #1: **Yes: **Frida Soesanti

Reviewer #2: No

---

## [Editor Report · Acceptance letter]

10 Nov 2021

PONE-D-21-14641R1 

Association between early gestation passive smoke exposure and neonatal size among self-reported non-smoking women by race/ethnicity: a cohort study 

Dear Dr. Grantz:

I'm pleased to inform you that your manuscript has been deemed suitable for publication in PLOS ONE. Congratulations! Your manuscript is now with our production department. 

Kind regards, 

on behalf of

Dr. Sze Yan Liu 

Academic Editor

PLOS ONE